

# How caged salmon respond to waves depends on time of day and currents

Ása Johannesen[1], Øystein Patursson[1,2], Jóhannus Kristmundsson[1], Signar Pæturssonur Dam[3] and Pascal Klebert[4]

[1] Department of Technology and Environment, Fiskaaling, Hvalvík, Faroe Islands
[2] RAO, Kirkjubøur, Faroe Islands
[3] Hiddenfjord, Sørvágur, Faroe Islands
[4] Seafood Technology, Sintef OCEAN, Trondheim, Norway

Corresponding author
Ása Johannesen, asajoh@fiskaaling.fo

## ABSTRACT

Disease, pest control, and environmental factors such as water quality and carrying capacity limit growth of salmon production in existing farm areas. One way to circumvent such problems is to move production into more exposed locations with greater water exchange. Farming in exposed locations is better for the environment, but may carry unforeseen costs for the fish in those farms. Currents may be too strong, and waves may be too large with a negative impact on growth and profit for farmers and on fish welfare. This study employed two major fish monitoring methods to determine the ability of Atlantic Salmon (*Salmo salar*) to cope with wavy conditions in exposed farms. Echosounders were used to determine vertical distribution and horizontal preference of fish during different wave and current conditions as well as times of day. Video cameras were used to monitor shoal cohesion, swimming effort, and fish prevalence in locations of interest. The results indicate complex interacting effects of wave parameters, currents, and time of day on fish behaviour and vertical distribution. During the day, hydrodynamic conditions had stronger effects on vertical distribution than during the night. In weak currents, fish generally moved further down in taller waves, but stronger currents generally caused fish to move upwards regardless of wave conditions. Long period waves had unpredictable effects on vertical distribution with fish sometimes seeking deeper water and other times moving up to shallower water. It is unclear how much the cage bottom restricted vertical distribution and whether movement upwards in the water columns was related to cage deformation. In extreme cases, waves can reach below the bottom of a salmon cage, preventing fish from moving below the waves and cage deformation could exacerbate this situation. Farmers ought to take into consideration the many interacting effects on salmon behaviour within a cage as well as the potential for cage deformation when they design their farms for highly exposed locations. This will ensure that salmon are able to cope when storms and strong currents hit at the same time.

# INTRODUCTION

Globally, fish is a major contributor of animal protein for human consumption, representing 17% of animal protein consumed in 2015 (*FAO, 2018*). Of all aquatic
animals produced globally, aquaculture accounts for 44% and has shown steady growth for two decades whereas capture fisheries have been stable. Atlantic salmon (*Salmo salar*) and other salmonids are amongst the most valuable species in aquaculture with growing demand. Due to the high value and increasing demand, there is understandable incentive for industry to expand production. However, most suitable salmon farming areas are already close to fully exploited as seen from waning production growth compared to other farmed species (*FAO, 2018*), so a necessity has arisen for exploring less convenient farming areas.

One other reason for the need for change is that there are highly destructive parasites affecting most marine based salmonid production. The copepod parasites sea lice and salmon lice (*Caligus spp* and *Lepeophtheiris salmonis* respectively) attach to salmon and consume salmon mucous, blood, and skin. In large numbers, these parasites can cause injury and osmotic stress, in extreme cases leading to mortality (*Johnson et al., 2004*). The cost of keeping salmon parasite free is great, but the costs in both fish welfare and monetary value of not doing so, are greater (*Costello, 2009*; *Liu & Bjelland, 2014*). In deep fjords, salmon can be manipulated so that they don't come into contact with surface water containing infectious stages of salmon lice. This is done either by submerging cages or using lights (*Hevrøy et al., 2003*) and feeding (*Frenzl et al., 2014*). Some submerged cages include a "snorkel", which is a vertical tunnel providing access for salmon to the surface without exposing them to the surrounding surface waters (*Oppedal et al., 2017*). Other methods of avoiding parasite infection include the use of skirts and cleaner fish (*Imsland et al., 2014*; *Frank et al., 2015*). Transitioning away from sheltered sites with low currents may decrease infection pressure due to the more rapid dispersal of infectious sea lice in locations with strong currents (*Kragesteen et al., 2018*). An additional benefit of farming in exposed locations is that water quality is usually better with stronger currents surrounding the farm providing higher oxygen saturation and elevated water exchange.

In strong currents, salmon spend more energy swimming against the current. This is likely to affect their growth and if currents are too strong also their welfare (*Johansson et al., 2014*). Additionally, in more exposed areas, waves are likely to be larger and the locations may be more often affected by bad weather conditions. In short choppy waves, there is a risk that salmon are unable to avoid collision with each other and the cage netting. This can, if movements are strong enough, lead to injury and poor welfare, but salmon ought to be able to move deeper in the cage to avoid these waves, as they attenuate quite quickly in the water column (Fig. S1). In long period swells, the salmon may not be able to escape the wave completely, as especially horizontal movement extends deep into the sea. In these cases, the risk to salmon welfare will depend on the extent to which the waves exceed the capability of the salmon to cope, and fortunately, water movement is slower in long period waves (Fig. S1). Salmon are good swimmers, but cage deformation in strong currents and large waves could limit movement and thus coping methods. *Klebert et al. (2015)* showed that flow velocity reduced by 40% and cage volumes decreased by 30% due to deformation under high currents in commercial sized circular cages. Flow hydrodynamics around and inside the cage affect the swimming speeds and schooling structures of fish (*Johansson et al., 2014*) and influence the distribution of feed in the cage during feeding. Optimal use

of feed is important for fish farmers as feed represents one of the greatest costs associated with fish farming (*Rocha Aponte & Tveterås, 2019*). In highly exposed locations, weather could affect feed consumption, potentially causing a risk that the salmon do not receive adequate nutrition. Prolonged underfeeding is both detrimental to welfare and to growth and monetary value for the farmer.

Salmon behaviour within salmon cages has been well documented in the literature. Vertical distribution is known to be affected by light and stratification in the water column, such as temperature or salinity differences. When there is enough light so that salmon are able to easily see each other and their surroundings, they need less space to avoid collision. In daylight they are also likely to move deeper down in the water column if not influenced by a thermocline or other factors (*Oppedal et al., 2001*; *Oppedal, Dempster & Stien, 2011*; *Hedger et al., 2017*). A variety of techniques to monitor salmon behaviour within a cage have been developed. These include echolocation, telemetry tags (data storage tags and transmitting tags), and video recording. To a lesser extent, PIT (RFID) tags with multiple receivers have also been used (*Nilsson et al., 2013*). Telemetry tags are very useful for obtaining precise information about individual fish such as heart rate, temperature, and depth. Tags can also be equipped with accelerometers. However, a recent study has found that there are welfare implications with tags affecting salmon buoyancy (*Wright et al., 2019*). If salmon are unable to correct for effects on buoyancy, for instance if they are in submerged cages, salmon may need to spend more effort on swimming or finding a suboptimal depth to counter the altered buoyancy. When there is a real chance that salmon will choose to move to greater depth to avoid waves, monitoring the fish using equipment that directly affects their depth preference can produce erroneous results. Echo sounders have been used for several years to monitor location of salmon within a cage. These are useful for monitoring shoal depth and biomass above the echo sounder. The information gained from echo sounders is not suitable for monitoring individuals but can provide useful information on general shoal positioning within the cage and swim bladder air content (*Glaropoulos et al., 2019*). Video recordings are limited to small areas within the cage and in the number of individuals that can be observed. There is risk of selection bias in terms of the individuals observed, as it is possible that certain salmon prefer the specific location where the camera is attached, be it depth or horizontal position. Cameras are often also limited to only monitoring fish during the day, which means that observations can be very limited in winters in the far north. Finally, despite recent developments in facial recognition (*Cermaq, 2018*), it is usually not possible to identify individual salmon in a video feed, so each observation must be treated as a single sample even considering selection bias. Even with such limitations, cameras provide information that is extremely difficult to get any other way: video cameras provide good information on swimming speed (BL s$^{-1}$, but care must be taken to account for currents), swimming effort (tail beats per minute), shoal cohesion and direction, and specific behaviours such as gill flaring or startle responses. If good cameras are available, it may be possible to observe breathing (opercular movement).

In this study, we monitor salmon behaviour in an exposed farm. Echo sounders are used to monitor vertical distribution and to examine differences in depth preference between

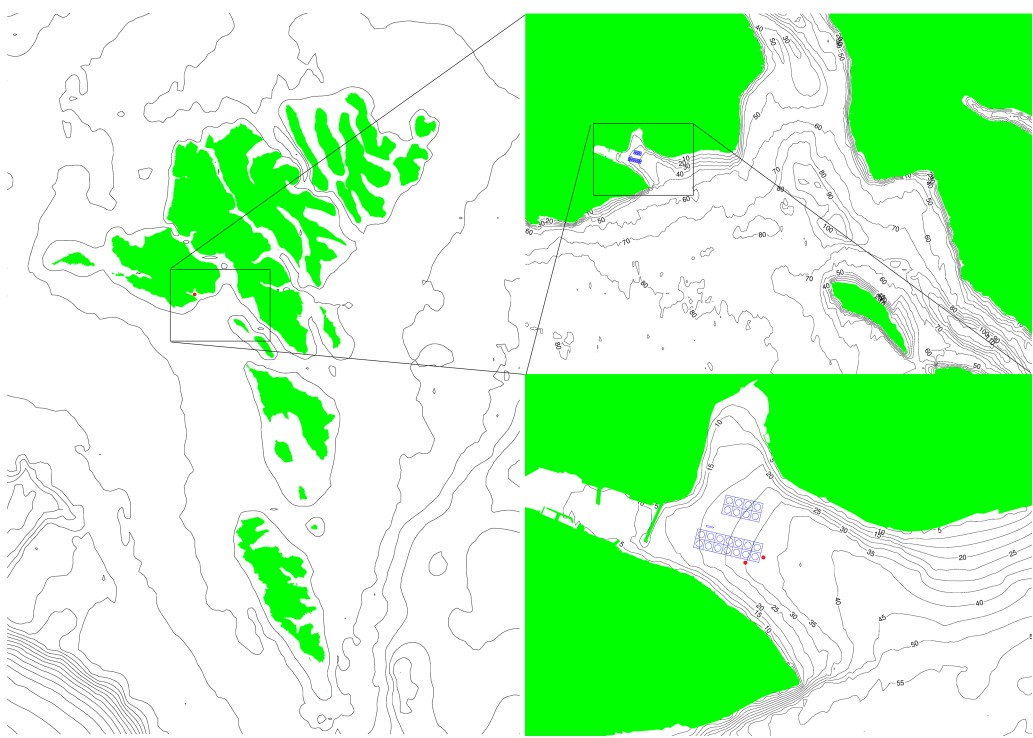

**Figure 1  A map of the Faroe Islands giving an overview of surrounding conditions around the farm and the locations of the two ADCPs (red dots).** The two images on the right are zoomed in from country level to area and finally to farm site to give an overview of how waves can enter the farm and the opportunities for reflections and refractions.

two locations in the cage. Video recordings are used to monitor shoal cohesion, proximity to the camera locations and swimming effort. These data are modelled against wave and current data to determine how wave height, wave period, current speed, and a combination of all three affect the behaviour of salmon.

## MATERIALS & METHODS

### Study area

Work was carried out at Hiddenfjord's Mi vágur farm on the Faroe Islands (62°02′33.7″N 7°09′27.2″W, Fig. 1). The salmon farming site in Mi vágur is a site that is highly exposed to waves, and to a much lesser extent to currents. The maximum wave height measured at the most exposed cages is 5.3 m (*Simonsen & Patursson, 2013*). The maximum current speed measured at the same location is 47 cm s$^{-1}$ (*Larsen et al., 2012*). The specific cage chosen for field work was located at the most exposed end of the farm, where waves moving into the fjord would hit the cage without any obstructions such as other cages (Fig. 1). The site has little to no stratification with similar temperature throughout the water column (Fig. S2).

The site is quite open to waves but is not facing in the direction from which the largest waves are coming. The largest waves are from southwest to west, while the bay is facing

southeast. The waves from the west are large ocean waves (*Niclasen & Simonsen, 2012*) that access the area outside the bay almost unhindered. The waves that enter the bay are refracted around the southern point of the bay or reflected from the neighbouring islands. To the southeast, the bay is sheltered by the neighbouring islands. The waves from southeast are therefore a mix of locally generated waves from the area southeast of the bay and swell that has either travelled between or around the islands and will arrive from more southerly directions. From the above it is assumed that the wave directions entering the bay from both the south-easterly and south-westerly storms are a mix of directions that turn out to be quite similar. Inside the bay there is another reflection from the northeast side of the bay, generating a very complex wave situation.

There is a big difference in wave length depending on the origin of the waves. Waves from south-westerly storms generally have peak period (Tp) of 14–20 s (*Patursson, 2019*) and it will generally be the longer period waves that are refracted into the bay and reflected from neighbouring islands (e.g., *Holthuijsen, 2007*). Storms from south-easterly directions generally have Tp = 12–14 s (*Patursson, 2019*), and since the waves entering the site from these directions are a mix of swell and locally generated waves with even shorter periods, Tp on the site might be even shorter than that.

The site is generally exposed to complex waves, maybe even partially standing waves at times, with short choppy waves coming from south-easterly directions and long waves from south-westerly directions. The maximum wave height from south-westerly directions is assumed to be higher than from the south-easterly directions (*Simonsen & Patursson, 2013*).

The currents inside the bay where the fish farm is located are tidally driven and a circulation in the bay is driven by the tidal currents outside the bay. The rotation is approximately half a tidal cycle in each direction (6 hrs). The currents at the most exposed cages are strongest in the north-north-westerly directions during the clockwise rotation in the bay. The location of the cages is such that they do not experience the strong currents during the anti-clockwise rotation. The maximum tidal currents at this location are expected to be around 30 cm s$^{-1}$ (*Joensen, 2017*).

## Study animals

The cage was stocked with 150 tonnes of salmon with an average weight of 2.03 kg. The biomass at the time was 7.54 kg m$^{-3}$.

## Instrumentation and deployments

Two ADCPs (Acoustic Doppler Current Profilers) were used at the site: these were 5-beam 500 kHz (Sentinel V50) from RDI Teledyne. Both ADCP's were seabed frame deployed with a gimbal which insured that these were always pointing vertically. The sampling frequency was set to 2 Hz and data were collected from 2018-02-26 to 2018-03-28. Both current profilers were programmed to record current every 5 min for two minutes and waves once per hour for 20 min. The profilers were located such that they could be used to estimate the amount of current and waves that the monitored cage was exposed to. Due to their positioning outside of the cage (Fig. 1), these did not provide information on the
actual within cage water conditions, but drag is likely to decrease current strength within the cage compared to outside of it (*Winthereig-Rasmussen, Simonsen & Patursson, 2016*). All sensors had internal loggers and data were collected at the end of the field work period.

Two echo sounders (EK15 from SIMRAD, 26° viewing angle) were used for monitoring of vertical distribution of salmon. The echosounders were mounted underneath the net by a line from each echosounder to opposite points on the circumference of the bottom net as well as a line between the echosounders to maintain distance between the echosounders and position under the net (Fig. S1). Both echosounders were weighted down by a 10 kg weight and kept vertical by a gimbal system. The depth of the echo sounders was 30 m and the bottom of the cage reached approximately 15 m above the echo sounders. This means that at the bottom of the cage, the beam width was approximately 7 m in diameter and at the surface it was approximately 13.5 m wide. Because the echo sounders were placed at opposite sides of the cage, one near the "front" facing side where most waves entered the cage (T1), and one near the "back" side where waves exited the cage (T2), the data could be used to determine whether salmon showed any preference in terms of horizontal positioning in relation to wave direction.

Behaviour was recorded on video using four video cameras recording simultaneously. Two were attached to the sides of the cage facing horizontally into the cage, one at 4 metres depth and one at the bottom of the side net (10 metres depth). Another two were attached to the bottom of the cage looking up, one approximately half way between the side and centre of the cage and one adjacent to the side of the cage (Fig. S1). All cameras were located in the more exposed side of the cage. The cameras were set to record 15 min (three five-minute videos) every two hours throughout the day and footage during the time when it was too dark to record any useful information was excluded after the fact. Cameras quickly became over grown with algae, so the data were limited to a couple of weeks of footage (2018-03-31 to 2018-04-11).

## Procedure

The equipment was deployed concurrently, and data were collected for up to 2 months. Due to storage and battery limitations, sometimes the data stored were not concurrent. Video recording was stored using a cloud storage service while other data collection was secured in-situ in data loggers. Video recordings were monitored on good and bad weather days throughout the day to secure behavioural footage throughout the tidal cycle as well as the diurnal cycle.

## Data processing
### ADCP data

Velocity data obtained from ADCP measurements were recorded in earth coordinate format. To reduce the impact of uncertainties on the measurements, data were averaged over 240 samples (two minutes). Wave parameters were calculated for every hour from 20-minute wave bursts.

### Echo sounder data

Echo sounder data were extracted from raw files using EchoView (v 9.0.279.33861). Before export, EchoView's bottom finder algorithm was used to find the surface. Making the detected surface zero metres at any one time allowed for interpreting the data in relation to water surface rather than the variable position of the echo sounders. Rather than keeping individual pings, data were exported using the PRC Nautical Area Scattering Coefficient (PRC_NASC) with cells of 10 min by 20 cm depth interval. Echo sounder data were cleaned of noise by excluding values weaker than 10,000 $m^2nmi^{-2}$. For purposes of determining vertical distribution of fish, the five shallowest and deepest values at each time point were extracted from the echo sounder data. As this was done after the surface and noise values were removed, these values were taken to represent the upper and lower bounds of the fish seen in the echo sounder data, subsequently referred to as the "shoal". Data below 20 m depth were removed, as these could not possibly be within the cage, so were taken to be wild fish outside of the cage. The remaining values correspond well with the outline of the salmon shoal (see Fig. S3 for an example of the shoal this method outlined).

### Video data

On each day of recording, 15 min of video footage (consisting of three 5-minute videos) was recorded at a two-hour interval throughout the day. From each video, one frame was extracted at 60 s into the video. This was used for determining a) presence of a shoal, b) if a shoal was present, swimming direction, and c) a binary measure of whether there were "many" fish (determined as more than 100 fish within five metres of the camera) seen in the two cameras looking up from the bottom. This resulted in three frames for each 15-minute recording session. Fish shoaling was determined as a large majority of the fish seen (more than 80%) facing in the same direction at uniform distances from each other. In cases where no clear shoal could be defined, swimming direction could also not be defined as fish were not overwhelmingly swimming in the same direction. In addition to the information extracted from snapshots, each five-minute video was also used to determine tail beat speed for one fish, resulting in three measures of swimming effort per recording session. The videos were also used to determine whether the fish were moving in relation to the fixed camera or maintaining their position.

Current profiler data was recorded until the 28th of March and video recordings started on the 31st of March. Due to the lacking overlap between video recordings and current profiler data, video footage was used to visually classify conditions into three qualitative wave categories: "Small waves", where very little and fairly regular movement was seen in the footage, "Medium waves" where there was noticeable horizontal and vertical movement with some irregularity of movement, and "Large waves" where vertical and horizontal movement was rapid and large causing fish to move in and out of view as water moved in relation to the video camera. It is unclear whether the movement seen was due to vertical movement of the camera or the water at any given time, so classification is based on camera movement (that is, the cage net movement) in relation to the fish visible in the footage.

## Data analysis
### Echo sounder data
In order to determine whole shoal effects of waves and current, the outer bounds of the shoal, determined by the previously mentioned five shallowest and five deepest values at each time point ("upper" and "lower"), were analysed separately. It was clear from echograms and variance analysis that depth variance increased at night and decreased during the day. Linear models of upper bound and lower bound depth found a significant effect of time of day on shoal depths ("Front", upper bound: $F_{2,15092} < 0.001$, $P = 0.985$; lower bound: $F_{2,15092} = 113.56$, $P < 0.001$; "Back", upper bound: $F_{2,13999} = 199.06$, $P < 0.001$; lower bound: $F_{2,13999} = 39.955$, $P < 0.001$). Therefore, echo data were separated into day and night for further analysis. Linear models with echo depths as dependent variable and current speed, Hs (significant wave height), and Tp (wave period) as independent variables were used to determine the effect of currents and wave size on the upper and lower bounds of the salmon shoal, at the back and front echosounder, and during the day and night. An interaction term between all three independent variables was included, but dropped as appropriate until the minimal adequate model was found. Because shoal density varied a lot over time, a weighted model was used where echo strength (PRC_NASC) was used as weights. This ensured that data points with stronger echo, or more fish carried greater weight in the analysis than data with relatively weak strength.

### Video data
Wave size was saved as an ordered factor and used as the independent variable in a linear model of wave effect on tail beats per second. Binomial family generalized linear models were used to determine the effect of wave size on swimming direction, shoaling, and presence of "many" fish in the two upwards facing cameras.

### Software
Data were analysed in R (*R Core Team, 2018*) and plotted using ggplot2 (*Wickham, 2016*). Videos were analysed using Event (*Stillman, 0000*), Boris (*Friard & Gamba, 2016*), VLC (*VideoLAN, 2019*), and the add-on "time" (*Mederi, 2018*) for VLC. The velocity software package provided by Teledyne (*Teledyne RD Instruments, 2017*) was used to partly visualize and post process the data from ADCPs. Waves and current data were exported from this software and further analysed and plotted using MATLAB® (*The Math Works, 2019*).

# RESULTS
## Conditions at the field site during data collection
The waves measured between 0.109 and 2.86 m in significant wave height (Hs) with a mean Hs of $0.872 \pm 0.537$ (mean ± SD). Wave data from current profilers corresponded well with wind data collected by local weather stations in a 90-degree increment from East to South (90-179 degrees; $F_{1,166} = 191.7$, $P < 0.001$, Fig. S4). This indicates that the waves travelling in these directions were wind driven rather than ocean swell and corresponds well with the known conditions at the site.

Wave period measurements indicate that the waves were mostly wind driven short period waves with very few measurements of more than 14 s peak period (Fig. S5). The two

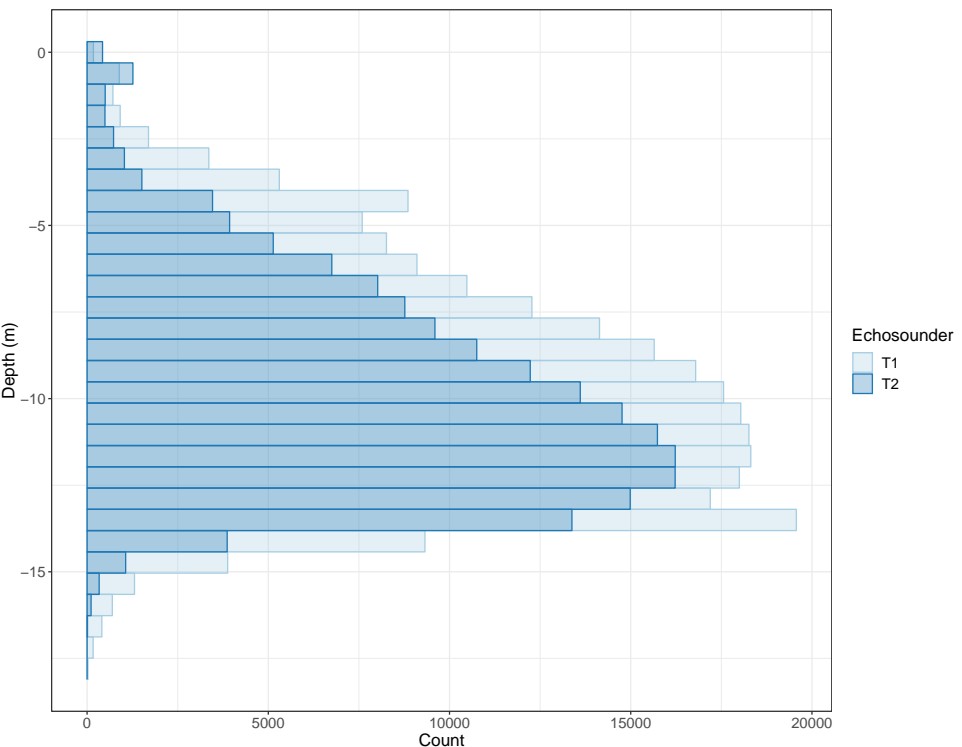

**Figure 2 Histogram of depths where PRC_NASC louder than 10,000 was recorded.** Counts of recorded PRC Nautical Area Scattering Coefficient (PRC_NASC) above 10,000 in each depth bin. The lighter colour is the "front" echo sounder (T1) and the darker colour is the "back" echo sounder (T2). Some echoes deeper than 20 m were removed as these were caused by fish outside of the cage.

instances of large waves with heights exceeding 2 m, the maximum peak wave period was 12 s. Currents were generally weak at less than 20 cm s$^{-1}$, but at the strongest tides, max current speed did exceed 20 cm s$^{-1}$ and peaked at 25 cm s$^{-1}$. Current direction changed between two main directions in a tidal pattern (Fig. S6).

## Echo data
### Overview
The salmon cage was 10 m deep at the sides with a sloping bottom resulting in the actual bottom of the cage being approximately 15 m deep at the point where the echo sounders were located (Fig. S1). Some cage deformation in connection with waves and current is to be expected in addition to some measurement error, so not all fish detected were above 15 m in depth, but most were (Fig. 2).

### Daytime behaviours
When the water was nearly still, with no current and no waves, the salmon maintained a depth between 9 and 14 m at the front and 10 and 13 m at the back of the cage. At the front, the upper bound of the salmon shoal moved up as current increased and this effect was exacerbated in large waves with a long period ($F_{7,6523} = 156.3$, $P < 0.001$, Fig. 3). The lower bound of the shoal also moved up with increasing current, so the shoal as a whole moved

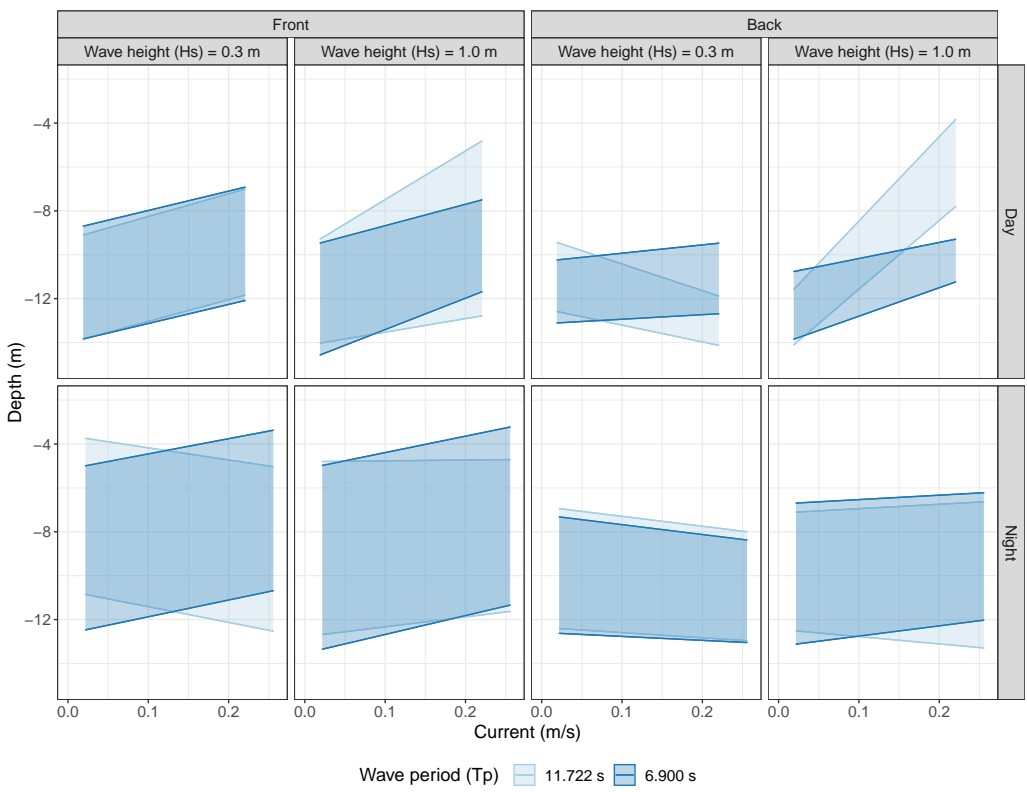

**Figure 3  Vertical shoal distribution over current speed at two wave heights and periods split into echo sounder location and time of day.** Shaded areas represent the predicted occupied vertical space as modelled by linear models using a three way interaction between current speed, wave height, and wave period. Data are separated horizontally into echo sounder position; "Front" and "Back" and wave height (mean Hs ±SD). Panels are vertically separated into time of day; "Day" and "Night". Shading colour represents long and short wave period (mean Tp ± SD).

up as current increased. In large waves and with strong currents, the shoal grew wider in long period waves and narrower in short period waves ($F_{7,6523} = 442.4$, $P < 0.001$). At the back, shoals were very narrow, spanning between two and four metres. Stronger currents generally caused fish to move slightly upwards in the water column. However, in long period waves increasing currents caused fish to move much higher up in the water column in large waves and down in small waves (Upper; $F_{7,5675} = 152.7$, $P < 0.001$, Lower; $F_{7,5675} = 287.1$, $P < 0.001$, Fig. 3).

### Night time behaviours

Shoals had a wider vertical distribution during the night (4–13 m at the front and 8-13 m at the back) in all conditions. At the front, fish were more widely dispersed in depth and shoals were generally nearer the surface. Fish mostly moved up in increasing currents and the shoal had a tendency to contract in large waves if wave period was long. In small waves, fish moved downwards with increasing current if wave period was also long (Upper; $F_{7,7050} = 50$, $P < 0.001$, Lower; $F_{7,7050} = 160.4$, $P < 0.001$, Fig. 3). At the back, fish moved slightly downwards when current increased and waves were small. In large waves, the shoals were

slightly wider and the upper bound of the shoals moved up in increasing currents. Wave period interacted with current speed such that shoals grew wider in stronger currents if wave period was long and slightly narrower if wave period was short (Upper; $F_{5,6894} = 58.13$, $P < 0.001$, Lower; $F_{7,6892} = 55.09$, $P < 0.001$, Fig. 3).

### Individual behavioural observations

Video data indicated a reduction in tail beat frequency with increasing wave size (Medium; $t = -2,273$, DF $= 332$, $P = 0.024$, Large; $t = -4.224$, DF $= 332$, $P < 0.001$, Fig. 4). In terms of other behaviours, frequency of fish shoaling was not significantly affected by wave height (Binomial GLM with ordinal predictor; Linear $z = 1.846$, $P = 0.065$; Quadratic $z = -0.335$, $P = 0.738$; $n = 325$). When fish did shoal and a general swimming direction could be ascertained, frequency of fish seen maintaining their position against the current decreased with increasing waves (Binomial GLM with ordinal predictor; Linear $z = -2.201$, $P = 0.028$; Quadratic $z = -0.333$, $P = 0.739$; $n = 325$, Fig. 5A) and swimming direction changed from being equally split to being predominately in one direction (Binomial GLM with ordinal predictor; Linear $z = -4.427$, $P < 0.001$; Quadratic $z = -0.490$, $P = 0.624$, $n = 238$, Fig. 5B). In the camera looking up from the bottom of the cage, there was no linear relationship between wave height and number of fish seen in the video, though there was a weak quadratic relationship (Binomial GLM with ordinal predictor; Linear $z = -0.055$, $P = 0.789$; Quadratic $z = -0.438$, $P = 0.031$, $n = 335$, Fig. 5C). The number of times where many fish were seen in the camera near the side of the cage decreased as wave size increased (Binomial GLM with ordinal predictor; Linear $z = -0.693$, $P = 0.005$; Quadratic $z = 0.013$, $P = 0.957$, $n = 260$, Fig. 5D).

## DISCUSSION

Waves and currents had interacting effects on salmon behaviour, which was further affected by time of day. In this study, there was no thermocline (Fig. S2), so changes in behaviour seen are likely to be directly caused by either hydrodynamic conditions or sunlight. Increasing current generally caused fish to move upwards and daylight generally caused fish to move downwards. Individual behavioural observations indicated that fish moved away from the sides of the cage in large waves and oriented their swimming according to waves, even overriding maintaining position against the current. Wave period interacted with other parameters in unpredictable ways. While short period waves had similar effects on fish as other parameters changed, long period waves caused fish to change their vertical distribution differently depending on horizontal location, current strength, and wave height. This effect of wave period is important to consider, because exposed farms are likely to be affected by long period waves more often than sheltered farms.

Although wave period in this study was never very long (Patursson, 2019), the effect on vertical distribution of salmon indicates that it had biological significance. In weak currents, wave period did not alter vertical distribution, but in strong currents, fish responded very differently in long period waves than they did in short period waves. During the day, fish moved upwards in tall long period waves and at the front, they dispersed widely in the water column. This upwards movement can be explained by an increased risk of collision

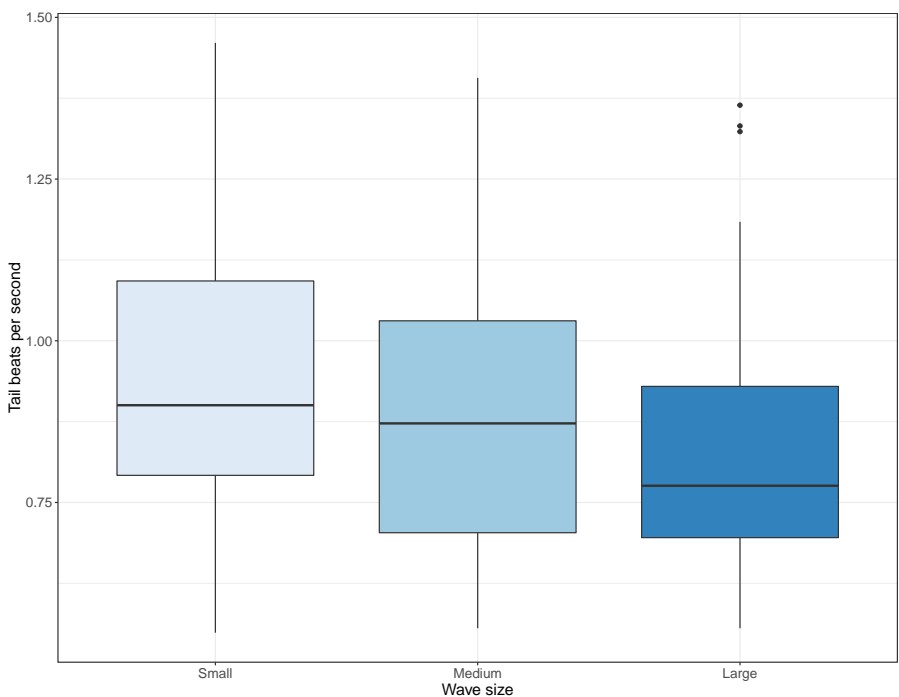

**Figure 4** **Distributions of tail beat frequency across three wave sizes registered from video recordings.** Tail beat frequencies (tail beats per second) observed in videos with small, medium, and large waves. Boxes and whiskers represent quartiles and dots are outliers.

for fish near the bottom of the cage as long period waves reach further down into the water column (*Dean & Dalrymple, 1991*). However, it does not explain why the fish moved downwards at the back of the cage when waves were not tall and also at night under various conditions. It is also possible that some of the behaviour seen was caused by cage deformation. The interacting effects of wave height and wave period on cage deformation in this highly complex wave situation are not well understood. While currents were so low that deformation due to currents alone ought to be limited (*Klebert et al., 2015*; *Gansel et al., 2018*), it is possible that this combined with waves could cause more cage deformation, mainly of the bottom net.

Shoals at night were generally more widely dispersed than during the day. The dispersal seen compared to the day was mostly in the upper bound of the shoal moving nearer the surface while the lower bound was only slightly shallower. This was not unexpected, as surface avoidance is less pronounced during the night (*Oppedal, Dempster & Stien, 2011*). Because of the wide vertical distribution during the night, changes due to hydrodynamic conditions were not as clear with fish always occupying most of the water column regardless of waves or currents. However, as fish were generally in shallower water, it is also possible that the fish were not as affected by cage deformation as they were not using the space near the bottom of the cage, even in calm conditions.

There was a general preference for being near the front of the cage, indicated by the generally bigger shoals there. The reason for this preference is unclear, but one

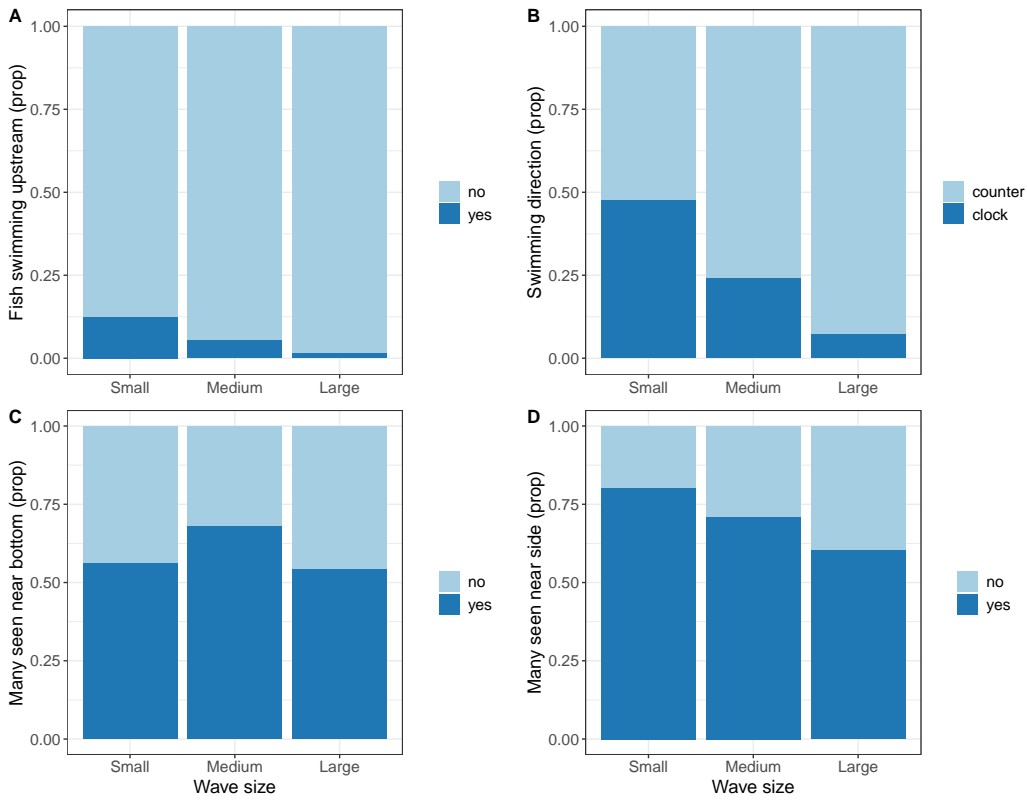

**Figure 5** Observations from video recordings expressed as proportion of extracted frames or video recordings where the observations are made. (A) Proportion of video recordings where fish were maintaining position against a current. (B) Proportion of extracted frames where a substantial majority of fish were facing either clockwise of anti-clockwise. (C) Proportion of extracted frames where many fish were seen in the bottom mounted camera. (D) Proportion of extracted frames where many fish were seen in the camera mounted near the side of the cage.

possible explanation is that fish tend to orient upstream in currents exceeding their preferred swimming speed (*Johansson et al., 2014*), and incoming current entered the cage approximately at the front, similarly to the waves. Whether currents were strong enough to cause this effect is unclear as the fish were larger in this study than in the study by *Johansson et al. (2014)* and the currents were weaker than the currents that caused the fish to change their swimming behaviour in that study. However, while the back of the cage is more sheltered from incoming current, it is slightly harder for the fish to navigate as they are facing away from the net, which serves as both a fixed point to navigate by and a collision risk. Therefore, the fish may have chosen to spend their time towards the front to avoid that collision risk. During the night, the difference between the front and the back was less pronounced, possibly due to a preference for wider dispersal causing the fish to use more of the horizontal space during the night.

Fish decreased their swimming effort in larger waves and spent less time maintaining position against the current. Observations from video data (Video S1) indicate that the tail beats were used more for balance than for propulsion in large waves. The camera used for

monitoring swimming direction was located such that fish seen swimming in a clockwise direction were moving towards where waves entered the cage. Therefore, the change in swimming direction to almost exclusively clockwise movement indicated a preference for facing towards large waves similar to when swimming in strong currents.

Overall, there are indications that especially during the day, physical spatial limitations were affecting the vertical distribution of fish. If cage deformation was limiting depth in the cage, then that explains the general upwards movement of fish in stronger currents. The bottom mounted camera positioned nearer the centre of the cage was intended to corroborate data from echo sounders in terms of shoal depth and there was an expectation of seeing more fish near the bottom mounted camera when conditions looked rough and fewer fish when conditions looked calm. However, there was no relationship between wave size and amount of fish seen near the bottom, indicating that fish did not move upwards in relation to the bottom of the cage. This indicates that any upwards movement of the shoal was dictated by the location of the cage bottom and not a change in preferred distance from the net.

Since cage deformation was not measured during this study, it is difficult to say with any certainty whether changes in vertical distribution of salmon were caused by cage deformation. More data on this would help provide more information on the behaviour of the salmon, especially at the bottom of the net. The bottom mounted camera data indicated that fish maintained their distance from the bottom regardless of waves, but wave classification was qualitative. The lack of effect of waves on proximity of fish to the bottom of the cage could be explained by the lacking detail in wave classification associated with the video data. Particularly the lacking information on wave period makes it difficult to compare video observations with echo sounder data. Future studies investigating swimming behaviour in relation to waves will benefit from greater detail by having concurrent wave and video data. Due to the limitations inherent in using video recordings to quantify behaviour, the behaviour seen in this study is only representative of behaviour performed in the camera locations. The locations were chosen to ensure that fish in key locations were observed based on previous knowledge of wave direction and salmon depth preference. However, it is possible that key information from other locations within the cage is missing.

## CONCLUSIONS

In general, behavioural needs of salmon change when hydrodynamic conditions change. There is a need to seek deeper water where waves are weaker, but there is also a need to move away from the net, which again may be encroaching on the available space if there is any net deformation. This decreases the available volume of the cage, so for the purposes of biomass, one ought to consider a cage exposed to large waves as having higher stocking density than one where there are no large waves. Both wave parameters and current speed need to be taken into account, as the effects of short period and long period waves as well as current speed all interact. Longer periods are generally associated with ocean waves whereas shorter periods are typically locally generated. In highly exposed sites, it is therefore likely that salmon will experience waves at great heights and with both short and long periods

and even more likely with combinations thereof. Also considering that currents can be expected to be strong at such locations, one ought to consider the spatial needs of salmon at the most extreme conditions and take into account cage deformation before deciding on stocking density and post-smolt size.

### Funding
The study presented in this article was carried out as part of research projects funded by the Norwegian Research Council: Environmental requirements and welfare indicators for new cage farming locations and systems (FutureWelfare: 267800/E40) and through the Centre for Research-Based Innovation on exposed aquaculture operations (SFI EXPOSED: 237790/O30) led by SINTEF OCEAN. This study was also financed by Fiskaaling through its own research grant program. The funders had no role in study design, data collection and analysis, decision to publish, or preparation of the manuscript.

### Grant Disclosures
The following grant information was disclosed by the authors:
Norwegian Research Council: 267800/E40.
Centre for Research-Based Innovation on exposed aquaculture operations (SFI EXPOSED): 237790/O30.
Fiskaaling.

### Competing Interests
Øystein Patursson is the owner of RAO, Kirkjubøur. Signar Pæturssonur Dam is employed by Hiddenfjord, and Pascal Klebert is employed by Sintef OCEAN.

### Author Contributions
- Ása Johannesen and Jóhannus Kristmundsson analyzed the data, prepared figures and/or tables, authored or reviewed drafts of the paper, and approved the final draft.
- Øystein Patursson and Pascal Klebert conceived and designed the experiments, performed the experiments, analyzed the data, prepared figures and/or tables, authored or reviewed drafts of the paper, and approved the final draft.
- Signar Pæturssonur Dam performed the experiments, authored or reviewed drafts of the paper, and approved the final draft.

### Field Study Permissions
The following information was supplied relating to field study approvals (i.e., approving body and any reference numbers):
Work was carried out at a fish farm belonging to Hiddenfjord with their permission.

### Data Availability
The raw data is available at Figshare: Johannesen, Asa; Klebert, Pascal; Kristmundsson, Jóhannus; Patursson, Øystein; Dam, Signar (2020): How caged salmon respond to waves

depends on time of day and currents. figshare. Dataset. https://doi.org/10.6084/m9.figshare.11406609.v1.

## Supplemental Information

Supplemental information for this article can be found online at http://dx.doi.org/10.7717/peerj.9313#supplemental-information.

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
