# Peer review of "How caged salmon respond to waves depends on time of day and currents"

_PeerJ, doi:10.7717/peerj.9313_

## Round 0.1 · original submission · Major Revisions

Your paper focusses in an important area of aquaculture research. Although It is scientifically sound the presentation needs to be deeply improved. This is specially evident in the discussion section when you have to compare your results with the available information.

Reviewer 1 ·

Basic reporting

No comments

Experimental design

Is it a limitation to one of the two methods applied for the evaluation of the objectives? Otherwise, please explain that acquired data were sufficient to extract the related results. At least, define entire time period that video recordings were acquired.

Validity of the findings

No comment

Additional comments

The aim of the study looks very interesting and provide state of the art knowledge on a very promising attempt to expand aquaculture towards a more sustainable way in the future.

In my opinion, manuscript should be accepted for publication, under some minor revisions.

Annotated reviews are not available for download in order to protect the identity of reviewers who chose to remain anonymous.

·

Basic reporting

Basic reporting

This manuscript addresses an important issue in salmon aquaculture, and presents interesting results based on a valid methodology, so has the potential for publication. However, the work needs be presented in a much clearer fashion. It requires a lot of polishing and more attention to detail.

1) The Discussion section could be a lot more focused.

For instance, a discussion could lead with the main points shown by the work (e.g. https://www.biosciencewriters.com/How-to-Write-a-Strong-Discussion-in-Scientific-Manuscripts.aspx), and then elaborate on these within the context of the literature (including a section on potential weaknesses/limitations). The discussion in this manuscript is not well organized. For instance, the sub-sections are “Daytime behaviour", “Night time behaviour" and “Video data” – the first two sections are on behavior, so surely the final section (“Video data”) should also be named in the same way? – for instance “Swimming behavior”

The authors open with a line (“When there is enough light so that salmon are able to easily see each other and their surroundings, they need less space to avoid collision.”) that feels like it could be part of the Introduction section. It is not a strong opening. What about a paragraph summarizing the key findings of the research, followed by an expansion on these in subsequent sub-sections?

The authors often use the present tense to describe their results in the Discussion section, which implies that that they are talking in general, when they are sometimes clearly referring to their own work. For example, L409 “In strong currents, fish mostly move downwards as opposed to during the day with the exception of fish at the front of the cage in short period waves.” Or L460 “There is a need to seek deeper water where waves are weaker, but there is also a need to move away from the net, which again may be encroaching on the usual space if there is any net deformation.” I strongly suggest that they change to pass tense except when the authors talking in general.

A lot of the Discussion reads a bit like the Results section. For instance, a lot of the text on daytime and night time behavior is just repeating what was in the Results section.

The Discussion is also sometimes a bit repetitive. For instance, the final paragraph on video data begins with “This indicates a preference for avoiding the net and orienting according to wave rather than current when waves are large.” (i.e. begins with net avoidance in large waves) and ends with “This makes sense considering the increased risk of collision with the net that is likely when waves are large and unpredictable.” (i.e. ends with net avoidance and large waves). It could be much more succinct.

The Discussion is also quite difficult to read, with a lot of description of results that could be summarized better. The authors could consider introducing each paragraph in the Discussion with a topic sentence, which summarizes what the paragraph is about. For instance, what about merging the “Daytime behaviours” and “Night time behaviours” sub-sections, and introducing the text with a clear statement (“Fish were more spread out during night than day”), and then expanding upon this? Framing with an introductory topic sentence makes it easier for the reader.

2) There are too many figures.

In particular, figures dealing with shoal position and distribution (Figure 8, 9, 10 and 11) are difficult to compare with one another because they are separated across multiple pages. Could these be integrated into one figure (each figure would be a panel) as follows?
i. Upper left panel – front of cage, daytime
ii. Lower left panel – back of cage, nighttime
iii. Upper right panel – front of cage, daytime
iv. Lower right panel – back of cage, nighttime
This would allow the reader to directly compare between the front and the back of the cage, and between daytime and nighttime

Figures 1, 4, 5 and 6 could potentially be in the supplementary material.

3) The English is generally ok, but could be a bit more formal and precise.

The authors should use past tense when describing their results (and when referring to them in the Discussion). Sentences could be made more direct, with fewer clauses. I go into more detail in General Comments below.

Experimental design

I can find no big problems with the experimental design. However, the authors could be a bit clearer about the statistical methods.

1) How did they find the upper and lower bound to be used in the linear models?

2) The authors need to also include some text of how they qualitatively chose wave size for analysis of the video data (first addressed in the Discussion on L449)

Validity of the findings

The findings are valid

Additional comments

1) The abstract needs improvement.

There are seven introductory sentences before the authors get to their study, which is too long.

Also "This study employs two major fish monitoring methods to determine the ability of Atlantic Salmon (Salmo salar) to cope with wavy conditions”. Add “…in offshore fish farms” (otherwise it is too general).

“Video cameras are used to monitor … proximity to camera locations” Would “spatial distribution” be better than “proximity to cameras” (the study is not interested in proximity to camera locations per se).

“The results indicate interacting effects between wave parameters and currents.”. So one would expect the following sentence to lead on from this (i.e. on interaction between waves and currents), but it then talks about fish behavior instead.

“In weak currents, fish generally move further down in taller waves, but stronger currents generally caused fish to move upwards.” This has too many clauses and is difficult to read. Change to “Fish generally moved downwards under conditions of weak currents and taller waves, …”?

The abstract does not mention day night effects, which is a key part of the findings.

2) The authors should make sure they are presenting scientific units correctly. E.g. write “14-20 s” rather than “14-20sec”, “30 cm s-1” rather than “30cm/s”

3) Regarding sub-headings in Material and Methods, could the opening paragraph and the sub-sections “Waves” and “Currents” all merged into the same subsection? e.g. “Study Area” (because they are all describing the study area).

4) I suggest changing the order of the figure panels so that they are consistent with the order of the text (L338-350). e.g. they are referred to in the text in the following order: Fig 13 D, Fig 13 A, Fig 13 C, Fig 13 B, whereas it should be A, B, C and then D. Also, the text on L340 implies that Figure 13 only refer to then fish were shoaling (“but when fish did shoal”)? If so this should be mentioned in the figure caption.

5) In the Materials and Method section, I suggest moving the video camera text (L179-186 and L224-239) to be after the echo sounder text, so that it is in the same order to that where it is presented in the results.

6) The figures could be improved.

Font sizes need to be increased.
Numbers should not be engineering notation. Axes should be capitalized (e.g. “Mar 04”, “count”).
Figure 1: the axes need labels, and the caption should specify “wave period (Tp)”. Tp (not T) should be used in the figures for the sake of consistency.
Figure 3: Can a scale be included? Or could this be superimposed onto Figure 1.
Legend for plots of shoal position and distribution: Suggest in full “Wave height (Hs)” “Wave period (Tp”).

7) Minor points

There are quite a few small changes that could improve clarity, including:
L45: “44 %”?
L69: remove comma after “there is a risk”
L93: should be “a recent study has found”
L122: specify the country
L124 better as “The maximum wave height measured at the most exposed cages is 5.3 m”.
L127: remove “the one”
L155: remove italics from “coming”
L175: “these did not”, not “these do not”
L225: “at a two hour interval”, not «with…»?
L321: “were distributed nearer to the surface” or “were distributed higher within the water column” better than “were generally higher up”
L334: Could be better expressed: “Video data indicated a reduction in tail bear frequency with increasing wave size”
L359 “muddies the waters” is too informal. “makes interpreting the data more difficult”?
L369 “cast more light” is too informal. “provide more information”?
L397: “However, the number of fish in the shoal does not explain the depth change as all of the shoal at the back is above 8m depth in large, long period waves and strong currents.” Not very clear and needs rephrasing.

---

## Round 0.2 · Minor Revisions

Thank you for improving your manuscript. However, some minor improvements to some figures could be made (see comments from reviewer 2).

Reviewer 1 ·

Basic reporting

Nothing to add

Experimental design

Nothing to add

Validity of the findings

Nothing to add

Additional comments

Nothing to add

·

Basic reporting

The reporting is now clear. The results and discussion have been improved. Some minor improvements to some figures could be made (see General comments for the author)

Experimental design

The experimental design is ok. The authors acknowledge potential limitations (mistiming between ADDP and video data, and their lack of information on cage deformation). Other aspects are fine, and are discussed well.

Validity of the findings

The findings are valid.

Additional comments

This manuscript has been greatly improved. The authors have done a good job in presenting the results, and the discussion is now a lot clearer and better structured. It presents interesting and useful findings, so I can recommend this for publication.

However, I can’t find a list of captions for the supplementary figures. Were these lost on the upload? I think these will have to be checked, so this part of the manuscript might require minor changes that probably do not require re-review.

There are a few (mainly optional) minor points that could be addressed, and some figures could be improved.

Minor points

L145: “(here meaning the directions between east and south)”. Probably not necessary. Remove?

L304: “Data are presented separately for T1 and T2 (“front” and “back”) echo sounders.” I don’t think this is necessary. Remove?

The authors should check that there is always a space between value and unit. e.g. “10 m”, not “10m”.

A very minor point: the final sentence of the conclusion uses “smolt”. Is this appropriate? Is “post-smolt” a better term for a salmon in a fjord, with “smolt” referring to the fish in freshwater?

Figures

Fig 2: the caption could include “PRC Nautical Area Scattering Coefficient (PRC_NASC)”.

Fig 5B should be same size as the other panels.

Fig S2 should have “Temperature (oC)” under upper panel. The x-axis values should ideally have the same precision (e.g. not 7.60 and 7.7 together).

---

## Round 0.3 · accepted · Accept

Dear Authors,
I am pleased to confirm that your paper has been accepted for publication in PeerJ.
Thank you for submitting your work to this journal.